# Efficacy of White Spot Syndrome Virus Protein VP28-Expressing *Chlorella vulgaris* as an Oral Vaccine for Shrimp

**DOI:** 10.3390/v15102010

**Published:** 2023-09-27

**Authors:** Min-Jeong Kim, Su-Hyun Kim, Jong-Oh Kim, Taek-Kyun Lee, In-Kwon Jang, Tae-Jin Choi

**Affiliations:** 1Department of Microbiology, School of Marine and Fisheries Sciences, Pukyong National University, Busan 48513, Republic of Korea; rnfma00082@naver.com (M.-J.K.); olem1140@gmail.com (S.-H.K.); jokim@pknu.ac.kr (J.-O.K.); 2South Sea Environment Research Division, Korea Institute of Ocean Science & Technology, Geoje-si 53201, Republic of Korea; tklee@kiost.ac.kr; 3Junggyeom Co., Ltd., Goyang-si 10223, Republic of Korea; jangik2001@gmail.com

**Keywords:** WSSV, VP28, vaccine, *Chlorella vulgaris*, immunization

## Abstract

The white spot syndrome virus (WSSV) is the causative agent of white spot disease, which kills shrimp within a few days of infection. Although WSSV has a mortality rate of almost 100% and poses a serious threat to the shrimp farming industry, strategies for its prevention and treatment are extremely limited. In this study, we examined the efficacy of VP28, a recombinant WSSV protein expressed in *Chlorella vulgaris* (*C. vulgaris*), as an oral shrimp vaccine. When compared with the control group, in which WSSV had a cumulative mortality of 100%, shrimp treated with 5% VP28-expressing *C. vulgaris* in their feed only had a 20% cumulative mortality rate 12 days after the WSSV challenge. When compared with the nonvaccinated group, the transcription of anti-lipopolysaccharide factor, C-type lectin, and prophenoloxidase genes, which are involved in shrimp defense against WSSV infection, was upregulated 29.6 fold, 15.4 fold, and 11.5 fold, respectively. These findings highlight *C*. *vulgaris* as a potential host for industrial shrimp vaccine production.

## 1. Introduction

Global shrimp farming is an important source of seafood, with the rise in demand every year leading to the production of about 5.51 million tons of shrimp in 2019 [1]. Whiteleg shrimp (*Litopenaeus vannamei*), the most farmed species of penaeid shrimp worldwide [1,2], is mainly produced in Thailand, Vietnam, and China. However, shrimp farming faces significant risks because of the threat of bacterial, fungal, protozoan, and viral infections [3], with viruses posing a particularly high risk. Currently, the main viruses of concern are the white spot syndrome virus (WSSV) and the yellow-head virus [4]. WSSV, which was first discovered in 1992 on shrimp farms in Asia, causes white spot disease (WSD), the most common disease in shrimp farms worldwide and a major cause of high economic losses [5].

WSSV-infected shrimp typically die within seven days, at a mortality rate of almost 100% [6]. The genome of WSSV, a circular, double-stranded DNA virus of the genus *Whispovirus* (family *Nimaviridae*) [7], is about 300 kilobase pairs long, but the genome size varies across isolates. WSSV is a bacilliform, nonoccluded, enveloped virus made up of a tail and a virion (210–380 nm long and 70–167 nm wide). Of the more than 40 characterized WSSV proteins, 21 are found in the envelope, 5 in the tegument, and 10 in the nucleocapsid [8]. WSSV’s viral proteins (VP) 15, VP19, VP24, VP26, and VP28 [8] play critical roles during infection.

To date, there are no effective ways of treating or preventing the highly infectious WSSV, probably because shrimp lack adaptive immunity [9]. Nevertheless, protection against WSSV is possible through the recognition of its structural proteins, VP19 and VP28, by the shrimp immune system [10]. For instance, the recombinant version of VP28, an envelope protein that plays a key role in WSSV infection, is thought to protect against WSSV [11]. Therefore, WSSV vaccines, such as an RNA nanovaccine and an inactivated vaccine, have been proposed [5,12]. However, considering the size of shrimp and large scale of shrimp culture farms, vaccination of individual shrimp is impossible and oral vaccines provided in a mix with feed is most desirable, as described elsewhere [12].

*Chlorella* is a genus of green, photosynthetic microalgae that exhibits rapid growth. Because *Chlorella* species are easy to culture, only requiring the supply of simple carbon and energy [13], they can yield high levels of protein under efficient, controlled culture systems [14]. *Chlorella*, which received “generally recognized as safe” certification from the United States Food and Drug Administration, is used as a shrimp feed additive as a substitute for fish meal [15]. In addition, the rigid cell wall of microalgae can encapsulate the expressed recombinant proteins and protect them from harsh conditions of low pH and digestive enzymes, thereby inducing both the systemic and mucosal immune systems. Because of these advantages, microalgae was suggested as a possible delivery system for the vaccine against the SARS-CoV-2 S-glycoprotein or the soluble form of the soluble ACE2 receptor as a decoy bind for the S-glycoprotein [16].

In this study, an oral WSSV vaccine using recombinant Chlorella that expressed WSSV’s VP28 was developed and evaluated. This oral vaccine has the potential to positively impact the shrimp farming industry.

## 2. Materials and Methods

### 2.1. Chlorella vulgaris (C. vulgaris) PKVL7422 Growth Conditions

The *C. vulgaris* strain PKVL7422 used in this study was isolated in our laboratory and deposited with the Korean Collection for Type Cultures (KCTC 13361BP). *C. vulgaris* PKVL7422 was cultured in a modified BG11 medium (BG11 supplemented with glucose at 0.5% *w*/*v*) under light at a photon flux density of 50 μmol photons m^−2^ s^−1^ and a temperature of 20 °C [17].

### 2.2. Construction of the VP28 Expression Vector

A VP28 gene (GenBank no. AY422228.1) with codon optimization for *C*. *vulgaris* was synthesized by Bioneer (Daejeon, Republic of Korea). The VP28 expression plasmid, pCCWVP28, was constructed by modifying a previously developed *Chlorella* transformation vector, pCCVG (Figure 1a) [17]. The pCCVG plasmid and the insert (the VP28 fragment) were digested using *BamH*Ⅰ and *Xho*Ⅰ restriction enzymes and then ligated to generate the pCCWVP28 plasmid. The pCCWVP28 plasmid has two flanking sequences (1000 bp long) derived from the *NR* gene of *C*. *vulgaris* PKVL7422, which enable its integration through homologous recombination, encompassing the cauliflower mosaic virus 35S promoter, the VP28 gene, and a transcription terminator from the Rbcs2 gene of *Chlamydomonas reinhardtii*.

### 2.3. C. vulgaris Transformation

*C*. *vulgaris* PKVL7422 was transformed through electroporation with two insert DNA fragments for triple homologous recombination, as described previously [17]. First, two PCR products with an overlapping sequence of 876 bp were produced using the primers 1-NR and 2-NR (Table 1, Figure 1b) using the following PCR conditions: 1 cycle at 95 °C for 5 min (pre-denaturation), 35 cycles at 95 ℃ for 20 s (denaturation), 60 °C for 20 s (annealing), and 72 °C for 4 min (extension), followed by a final extension at 72 °C for 10 min. A total of 5 × 10^7^ cells/mL (1 mL) were collected by centrifugation at 2600× *g* for 10 min. The pellets were then stabilized with 1 mL of osmosis buffer at room temperature for 1 h, followed by centrifugation. The pellets were then resuspended in 400 µL of electroporation buffer, followed by the addition of 2 µg of each PCR fragment. The mixture was then incubated on ice for 10 min before being transferred into a 0.2 cm electroporation cuvette, followed by electroporation using a Gene Pulse II Electroporation System (Bio-Rad, Hercules, CA, USA) at 1.0 kV, 25 μF, and 400 Ω. To allow recovery, the pulsed cells were seeded into 6-well plates containing 5 mL of BGNK broth [17] and incubated at 20 °C for 18 h in the dark. The cells were spread on BGNK agar plates containing 200 mM KClO_3_ and incubated at 20 °C for 14 days in the dark.

### 2.4. DNA Extraction and PCR Confirmation

Ten transformed *C*. *vulgaris* colonies were randomly selected and cultured in 12-well plates containing 2 mL of BGPK medium [17] under light at 20 °C for 7 days. Total DNA was extracted from 1 mL of the transformed cell culture, containing approximately 5 × 10^7^ cells, using the 5M Plant DNA Extraction Kit (Scinomics, Daejeon, Republic of Korea) according to manufacturer guidelines. The VP28 gene was amplified using VP28-specific primers (Table 1) [18]. The confirmed transformants were cultured in 20 mL of BGPK medium under light at 20 °C for 7 days.

### 2.5. Confirmation of VP28 Expression in Transformed C. vulgaris

To recover transformed *C*. *vulgaris*, 10 mL of culture medium containing 1 × 10^8^ cells/mL was centrifuged at 2600× *g* for 10 min. The supernatant was then discarded and the pellet lysed by incubation in 1 mL of RIPA buffer (BioBasic, Markham, ON, Canada) for 1 h on ice. Cells were then sonicated for 5 min (using 2 s pulses at 200 W) using a JY92-IIN sonicator (Ningbo Scientz, Ningbo, China). The supernatant was then centrifuged at 15,000× *g* for 1 min and the proteins resolved using 12% SDS–PAGE at 110 V for 100 min, followed by transfer onto polyvinylidene fluoride membranes. The membranes were then blocked using 5% skimmed milk in PBS–Tween 20 and then incubated with a primary antibody against VP28 (ab26935, Abcam, Cambridge, UK) and an HRP-conjugated anti-rabbit IgG(H+L) secondary antibody (PBio-Rad). The protein signal was then developed for 30 s using WESTSAVE STAR Western blotting reagents (Young in Frontier, Seoul, Republic of Korea) and imaged on an iBright CL1500 system (Thermo Fisher, Waltham, MA, USA). Expression levels were compared through density analysis using iBright analysis software, Desktop Version 5.2.0 (Thermo Fisher). Relative protein levels were calculated by normalizing the VP28 protein level to 1.

### 2.6. Shrimp

A total of 250 whiteleg shrimp (*L. vannamei*, mean length: 10.5 cm, mean weight: 4.5 g) were purchased from Chungsu Fisheries (Seosan, Republic of Korea). Ten shrimp were randomly selected and WSSV infection determined using nested PCR. To this end, 0.5 g of gastrointestinal and muscle tissue was placed in 2 mL screw tubes, followed by the addition of 1.5 mL of TNE buffer (50 mM Tris-HCl, 400 mM NaCl, 5 mM EDTA, 1 mM PMSF, pH 8.5). The tissues were then disrupted using a Bioprep-24 homogenizer (Scinomics) and then centrifuged at 15,000× *g* for 5 min to recover the supernatant. Total DNA was then extracted from the supernatant using a 5M Cell/Virus DNA extraction kit (Scinomics) using the manufacturer’s protocol, followed by nested PCR analysis using WSSV and nested WSSV primers (Table 1).

For the vaccine test, healthy shrimp (*n* = 180) were randomly divided into 4 groups of 15 each, including the positive control group, negative control group, wild type *Chlorella* group, and vaccine group. Each experiment (treatment) was performed in triplicate. To this end, groups of 15 shrimp were maintained in 50 L tanks with a supply of natural seawater and air at 28 °C. The water was exchanged daily with the introduction of 30 L of fresh seawater.

### 2.7. Immunization of L. vannamei

Wild type and transformed *C*. *vulgaris* (1 × 10^8^ cells/mL) were harvested by centrifugation at 3000× *g* for 10 min. Next, 20 g of the harvested cells was resuspended in 20 mL of PBS and sonicated using a JY92-IIN Sonicator (Ningbo Scientz) equipped with a 6 mm diameter tip and 400 W output for 10 min using 2 s pulses. Next, the homogenate was freeze-dried for three days using an FDU-2200 freeze-dryer (Sunil Eyela, Seongnam, Republic of Korea). A feed paste was then prepared by mixing 5 g of freeze-dried cells (transformant or wild type *Chlorella*) and 95 g of a commercial feed powder (KOFEC Co. Ltd., Naju, Republic of Korea) in 100 mL of distilled water. The same commercial feed was mixed with 100 mL of distilled water and used as the control feed. The feed mixtures were stored at −20 °C. After acclimatization to the experimental conditions for 3 days, the shrimp were fed three times daily (with a total of 1% of the average shrimp weight at each feeding) for 7 days. The control, wild type *Chlorella*, and vaccine groups were fed with the normal feed, wild type *Chlorella* feed, and vaccine feed, respectively.

### 2.8. WSSV Challenge

The WSSV-infected shrimp used as inoculum were obtained from the West Sea Fisheries Research Institute (Incheon, Republic of Korea). To determine the virus titer in the inoculum, muscles and gastrointestinal tracts were collected from five WSSV-infected shrimp, followed by DNA extraction from 0.1 g of muscle tissue and 0.1 g of gastrointestinal tract mixture using a 5M Cell/Virus DNA extraction kit (Scinomics) following the manufacturer’s protocol. The PCR product containing part of the VP28 gene was amplified using the qWSSV primers (Table 1) and used as a quantitative PCR standard. Quantitative PCR analysis was conducted using the extracted total DNA and qVP28 primers (Table 1), and the cycle threshold value was converted into copy numbers as described previously [19].

After immunization, a challenge test was conducted by feeding the WSSV-infected shrimp with a final virus titer of 1.56 × 10^7^ copies/g of feed. The wild type *Chlorella*-fed (WC) and transformed *Chlorella*-fed (TC) groups were fed thrice in one day, with 1% of the average shrimp weight per feeding. The negative control group was fed a normal commercial feed. After the challenge, shrimp mortality was measured over 14 days, and normal and vaccine feeds were continuously fed. Dead shrimp were examined for symptoms of WSSV infection, including typical white spots on the carapace. WSSV infection in symptomatic shrimp was confirmed using nested PCR.

### 2.9. Reverse-Transcription qPCR (RT-qPCR) Analysis of Immune-Related Genes

To confirm immune response induction by the oral vaccine, the transcription levels of genes involved in shrimp antiviral immune response were assessed. Total RNA was extracted from the shrimp hepatopancreatic region immediately after vaccination and then on days 7 and 14 after vaccination using a HiYield total RNA mini kit for tissues (RBS, Taipei, Taiwan) following the manufacturer’s protocol. Next, 1 µg of RNA was retrotranscribed using Reverse Transcription Premix (Elpis Biopharmaceuticals, Daejeon, Republic of Korea) followed by RT-qPCR analysis of anti-lipopolysaccharide factor (ALF), C-type lectin (CTL), and prophenoloxidase (proPO) using the primers shown in Table 1 and the following conditions: pre-denaturation at 95 °C for 1 min, followed by 40 cycles of denaturation at 94 °C for 15 s, annealing at 60 °C for 30 s, and extension at 70 °C for 30 s. β-actin was used as the reference gene. Relative gene expression was determined using the 2^−ΔΔ*C*t^ method [20].

### 2.10. Statistical Analyses

Statistical analyses were performed using two-way repeated-measures analysis of variance (ANOVA) on GraphPad Prism (GraphPad Software Version 8.2.1 San Diego, CA, USA) and multiple *t*-tests on Excel (Microsoft, Redmond, WA, USA). *p* < 0.001 indicates statistically significant differences.

## 3. Results

### 3.1. Selection of Transformed Chlorella vulgaris (C. vulgaris)

After 2 weeks of culture, colonies were observed on plates containing transformed *C*. *vulgaris* but not on those containing nontransformed, wild type *C*. *vulgaris* (Figure 2). PCR analysis on randomly selected transformed *C*. *vulgaris* colonies revealed products of the expected sizes. Western blot analysis revealed the expression of a 28 kDa protein in six randomly selected, transformed *Chlorella* samples but not in the nontransformed wild type *C*. *vulgaris*. A transformant named TF1, which expressed the highest level of the recombinant VP28 protein, was used in subsequent experiments.

### 3.2. PCR Analysis of the Inoculum and WSSV-Challenged Shrimp

Before shrimp vaccination and challenge with WSSV, 10 of the 250 shrimp used in this study were randomly selected and the presence of WSSV examined using standard and nested PCR. These analyses did not reveal PCR products of the expected band sizes (604 and 258 bp), which were detected in the positive control (Figure 3a,b). In the WSSV-infected shrimp used as feeding-based inoculum, qPCR analysis of muscle and gastrointestinal tract tissues revealed virus titers of 5.02 × 10^5^ and 6.08 × 10^5^ copies/mg, respectively (Figure 3c).

### 3.3. Efficacy of the Oral Vaccine

Mortality analysis revealed that in the positive control group that received normal feed, the final cumulative mortality on day 20 after the WSSV challenge was 100%, whereas no mortality was observed in the negative control (without the WSSV challenge) group (Figure 4). When compared with shrimp fed on wild type *C. vulgaris*, which had a final cumulative mortality of 77.8%, shrimp fed on VP28-expressing *C. vulgaris* had a final cumulative mortality of 20% and a relative percent survival rate (RPS) of 80% (*p* < 0.001, Figure 4a). All dead shrimp exhibited white spots on the inside surface of the carapace, which is a typical sign of WSD (Figure 4b). Moreover, WSSV infection was confirmed using PCR, which revealed the expected 258 bp products in DNA samples from all dead shrimp but not in DNA from the negative controls (Figure 4c).

### 3.4. RT-qPCR Analysis of Immune Response Genes

RT-qPCR analysis of the expression levels of proPO, ALF, and CTL using total RNA extracted from hepatopancreas tissues of vaccinated shrimp revealed that when compared with samples obtained from the positive control group immediately after being fed the normal feed, proPO, ALF, and CTL were significantly upregulated on days 7 and 14 after feeding (immunizing) the shrimp with the VP28-expressing *C. vulgaris* (Figure 5). The proPO gene was the highest expressed and was 29.6-fold and 28.2-fold higher on days 7 and 14 after immunization, respectively. On day 7 after immunization, the ALF and CTL levels increased 12.1 fold and 10.1 fold, respectively, whereas on day 14, their levels rose 15fold and 11.5 fold, respectively, when compared with the control group. In shrimp fed with wild type *C. vulgaris,* the transcription levels of proPO, ALF, and CTL were 1.2 fold, 3.2 fold, and 1.1 fold higher on day 7, respectively, and 1.0 fold, 2.9 fold, and 1.0 fold on day 14, respectively, when compared with the shrimp on the normal feed (*p* < 0.001, Figure 5).

## 4. Discussion

To date, there are no effective ways of preventing or treating viral infections in shrimp [21]. Nevertheless, various methods based on plants, seaweeds, nanoparticles, and probiotic bacteria [21] have been proposed for preventing WSSV infection.

VP28, a major WSSV envelope protein, plays an important role in the early stages of viral infection and has been widely used in the development of recombinant WSSV vaccines [22,23]. The injection of shrimp with recombinant VP28 proteins expressed using bacterial, algal, yeast, and plant transformation systems has exhibited protective effects against WSD [24]. Recombinant proteins in feed additives have also been shown to be effective as vaccines. For example, shrimp feed containing recombinant VP28-expressing *Brevibacillus brevis* exhibited a protective efficacy of up to 69% when compared with a survival rate of only 6.7–18.5% in shrimp fed with sonicated *Brevibacillus brevis* [25]. Moreover, shrimp feed coated with VP28-expressing *Bacillus subtilis* spores achieved an RPS of 83.3% [26]. However, before use as vaccines, recombinant proteins expressed in bacteria require purification to remove bacterial endotoxins [27,28]. Microalgae, such as *Chlamydomonas*, *Dunaliella*, *Phaeodactylum*, and *Chlorella*, which have “generally recognized as safe” certification [29], are used as eukaryotic expression vectors [30]. However, the plasmids used for microalgae transformation contain the desired genes as well as undesired antibiotic-resistant genes, whose risk of horizontal transfer to bacteria has raised concerns about the development of antimicrobial resistance [31]. The detection of several antibiotic-resistant bacterial strains and antibiotic-resistance genes in aquaculture settings has significantly reduced treatment options, resulting in significant losses [32]. To overcome these challenges, a *Chlorella* transformation system was used in this study, which used a DNA fragment without antibiotic-resistant genes to exclude antibiotic-resistant genes from the cloning vector (Figure 1) [17].

Our results demonstrate the effectiveness of a *C*. *vulgaris*-based oral vaccine against WSD. Although the theoretical size of the VP28 protein is 22 kDa, a 28 kDa protein was purified from WSSV, probably because of posttranslational modifications, such as glycosylation or phosphorylation [23,33]. In this study, Western blot analysis identified a 28 kDa protein, indicating possible posttranslational modification in *C*. *vulgaris* (Figure 2d). Posttranslational modifications, including glycosylation and phosphorylation, were reported in several microalgae species, including *C*. *vulgaris* [34,35].

Previous studies challenged shrimp with WSSV through injection [10,36], feeding [37,38,39], or immersion [40], and the time required for 100% mortality after the challenge depends on the inoculation method and virus titer. For example, Li et al. [36] reported 100% mortality within 5 days of intramuscularly injecting shrimp with 10^6^–10^7^ WSSV copies/mL, whereas McLean et al. [39] achieved 100% mortality within 10 days after feeding shrimp with 1.55 × 10^7^ WSSV copies/g. Because low inoculum levels make it difficult to evaluate vaccine efficacy, inoculum levels are typically adjusted to attain 100% mortality within 10–12 days after the challenge, although an exact virus titer threshold for this purpose has not been determined [40,41]. In this study, we administered 1.55 × 10^7^ WSSV copies/g of feed, which caused a cumulative mortality of 100% in the control group within 12 days after the inoculation (Figure 4a).

During vaccination, transformed *C*. *vulgaris* was added to 5% dry weight of the feed, which corresponded to a dose of 0.5 mg (wet weight) of transformed *C*. *vulgaris* per g of body weight in *L*. *vannamei*, and it resulted in a survival rate of 80% (Figure 3a). In contrast, Lanh et al. (2021) reported that feeding shrimp with VP28-expressing *Chlamydomonas reinhardtii* at 4 mg per g of body weight resulted in a survival rate of 70%. A reason for the higher survival of *L*. *vannamei* after consuming a lower dose of transformed *C*. *vulgaris* may be its high VP28 expression, which was estimated to be 11 mg VP28 per g (wet weight) of transformed *C*. *vulgaris*.

The subunit vaccine MBP-VP28, produced in *Escherichia coli*, resulted in an RPS of 44% when administered as an oral vaccine at a dose of 4 μg/g shrimp body weight [10]. In contrast, a VP28 DNA vaccine manufactured using *Escherichia coli* achieved survival rates of 52.5% and 56.6–90% when injected at 20 μg [36] and 30 μg [41] per shrimp, respectively. Oral vaccination has several advantages over injection, including lower production costs and lower stress on the shrimp. Although vaccine efficacy has been reported to vary with shrimp type, size, farming environment, and WSSV inoculation dose, *Chlorella* is a promising vaccine production platform.

Because shrimp lack adaptive immunity, cellular and humoral responses that mediate innate immunity are essential for their health [42]. Invertebrates generally rely on pattern recognition receptors (PRRs) that sense pathogen-associated molecular patterns (Figure 5d) [43]. In shrimp, cellular immune responses include phagocytosis, encapsulation, and melanization, whereas their humoral responses include complex enzyme cascades that control melanization. In addition, various defense enzymes, antimicrobial peptides, and antiviral peptides belonging to the Toll pathway are used for defense [42]. In response to WSSV infection, *L*. *vannamei* expresses antimicrobial peptides, such as ALF, alkaline phosphatase, and lysozyme, through the Toll pathway [42], as well as antiviral peptides such as CTL [44]. The induced ALF and CTL suppress infections by various pathogens, including *Vibrio alginolyticus* and WSSV [42,44]. VP28 is an envelope protein that is recognized by shrimp innate immunity, resulting in the upregulation of immune-related genes, such as proPO, ALF, and CTL [40].

The activation of PRRs, which detect and bind to pathogen-associated molecules, activates a proteinase cascade, the Toll pathway, and proPO-activating enzymes [45]. ALF and CTL, which are activated in the Toll pathway, are closely associated with WSSV infection. In *Crustacea* species, ALF exhibits broad and potent antibacterial activity and plays an important role in their defense against WSSV infection [46]. WSSV infection in ALF-knockout shrimp caused high mortality, which was significantly reduced by ALF supplementation [47]. Our data show that when compared with unvaccinated shrimp, ALF gene expression was increased 15.4-fold 14 days after oral vaccination with the VP28-expressing *C. vulgaris* (Figure 5a). CTL, which promotes phagocytosis and aggregation, exhibits antiviral activity [48]. Like ALF, treating WSSV-infected shrimp with CTL significantly reduced their mortality [49]. ALF and CTL levels are reported to increase within 4–36 h after WSSV infection [46,49], suggesting that WSSV-induced ALF and CTL expression are involved in protection against WSSV. In this study, CTL transcription levels increased 11.5-fold within 14 days after vaccination (Figure 5b), suggesting that as with WSSV infection, the administration of VP28 as an oral vaccine can induce an immune response in shrimp.

CTL also induces melanization in shrimp by activating the proPO system [50]. Melanization, which depends on proPO activation, is one of the fastest immune responses in invertebrates, including crustaceans [45]. After recognizing PRRs, proPO-activating enzymes activate proPO and convert it into phenoloxidase [45], which promotes melanization and the formation of a melanin polymer with pathogens in the shrimp’s shell, which is later eliminated through molting, resulting in a healthy shrimp [45]. In WSSV-infected *L*. *vannamei*, proPO is downregulated by WSSV’s immediate-early 1 (IE1) protein, which inhibits proPO activity. Infecting shrimp with IE1-silenced WSSV is associated with higher proPO expression levels [51]. In this study, shrimp that were orally vaccinated with WSSV’s VP28 expressed higher proPO levels (Figure 5c), which is consistent with previous findings that vaccinating shrimp with a subunit-feed vaccine containing VP28-expressing *Chlamydomonas* upregulated proPO [40]. This indicates that vaccination-induced proPO upregulation contributed to protection against WSSV (Figure 5c,e).

## 5. Conclusions

When administered as an oral feed-additive vaccine, VP28-expressing *C*. *vulgaris* PKVL7422 exhibited a strong protective effect against WSSV infection. Moreover, vaccinated shrimp expressed elevated levels of the genes proPO, ALF, and CTL, which are involved in shrimp defense against WSSV infection. Because *C*. *vulgaris* yielded high amounts of the vaccine protein VP28 and is associated with low production costs, ease of application, and high protective efficacy, it has great potential for use in industrial shrimp vaccine production, which may be a valuable tool for disease prevention in the shrimp aquaculture industry.

## Figures and Tables

**Figure 1 viruses-15-02010-f001:**
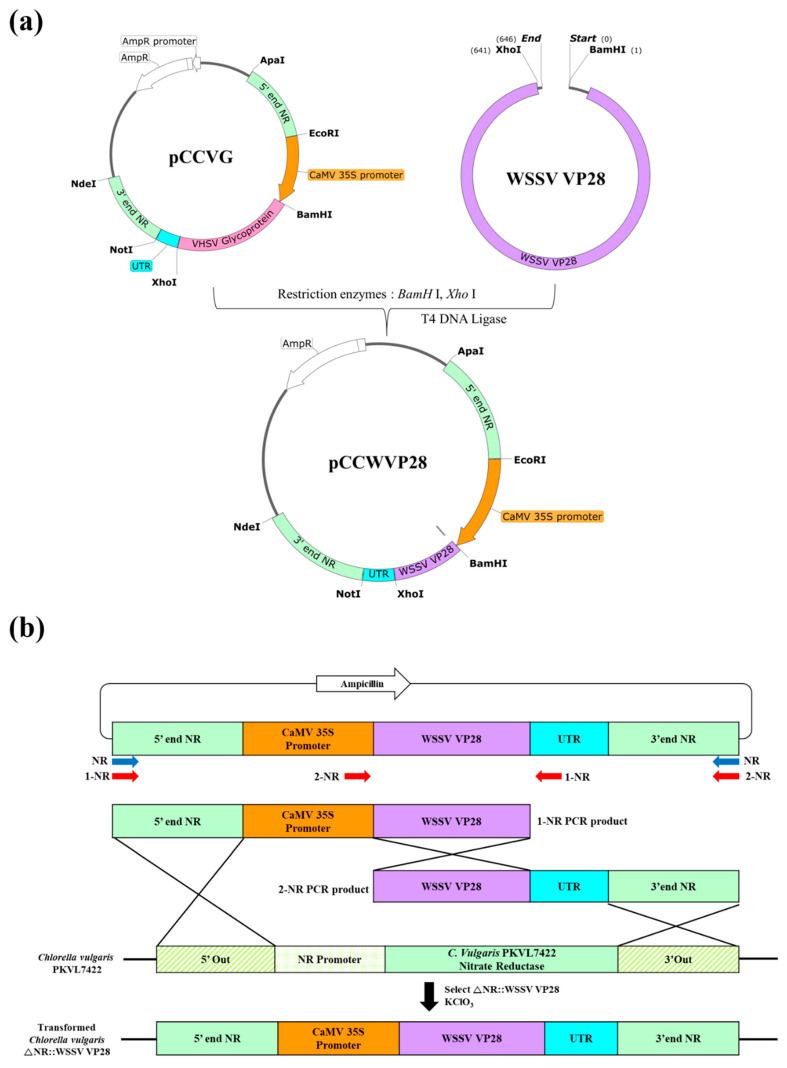
Construction and schematic representation of the structure of the pCCWVP28 plasmid for *Chlorella vulgaris* transformation. (**a**) The VHSV glycoprotein gene in the pCCVG plasmid was replaced with a codon-optimized VP28 gene to generate the pCCWVP28 plasmid. (**b**) The transformation was conducted using a split-NR transformation system with two insert DNA fragments. Three crossover events resulted in the replacement of the wild type NR gene with the DNA insert, which allowed the selection of transformants using KClO_3_.

**Figure 2 viruses-15-02010-f002:**
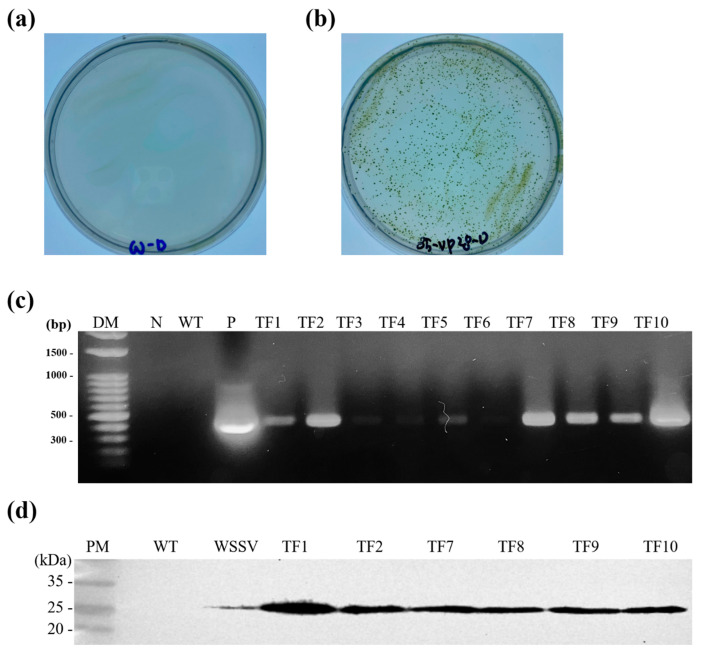
Selection of transformed *Chlorella vulgaris* (*C*. *vulgaris*) and confirmation using PCR and Western blotting analyses. (**a**) Wild type PKVL7422 on a BGNK plate containing 200 mM of KClO_3_. (**b**) Colonies of transformed *C*. *vulgaris* PKVL7422 on a BGNK plate containing 200 mM of KClO_3_. (**c**) PCR results of randomly selected transformed cells. Lane DM: DM3200 DNA marker (SMOBIO); lane N: negative control; lane WT: wild type *C*. *vulgaris*; lane P: positive control with pCCWVP28; lanes TF1–10: transformed *C*. *vulgaris*. (**d**) Western blot analysis of VP28 expression by selected transformants. Lane M: PM2700 protein marker (SMOBIO); lane WT: wild type *C*. *vulgaris*; lane WSSV: total protein from white spot syndrome virus (WSSV)-infected shrimp; lanes TF1, 2, 7, 8, 9, and 10: PCR identification of transformed *C*. *vulgaris*.

**Figure 3 viruses-15-02010-f003:**
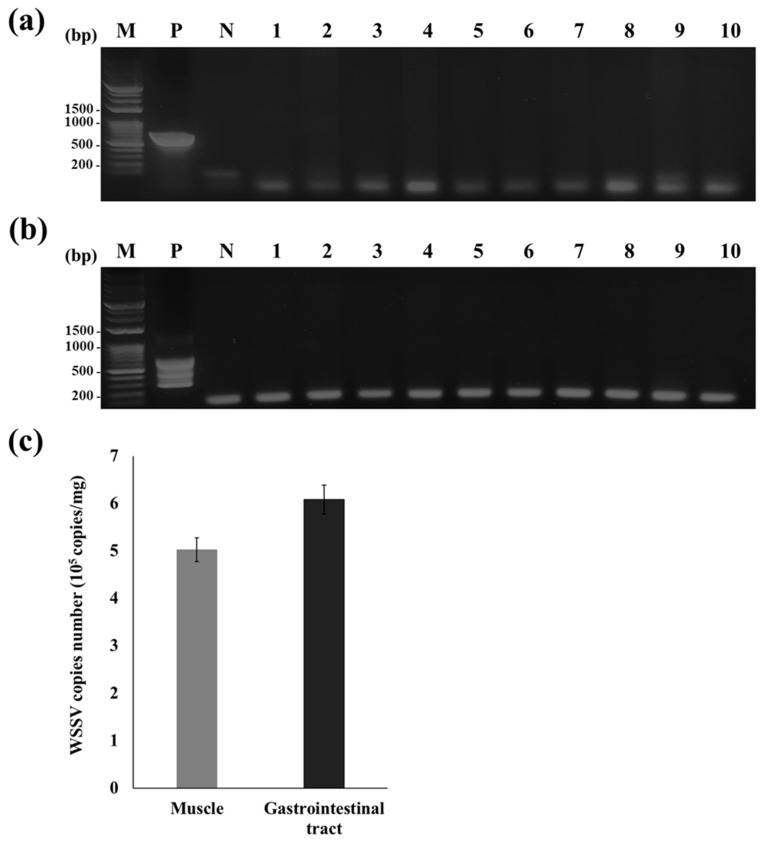
Presence of WSSV infection in test shrimp. (**a**) Confirmation of WSSV infection using standard PCR. (**b**) Confirmation of WSSV infection using nested PCR. Lane M: DM3200 DNA marker (SMOBIO); lane P: positive control (total DNA from *Litopenaeus vannamei* (*L. vannamei*) infected with WSSV); lane N: negative control; lanes 1–10: total DNA from the healthy *L. vannamei* used for the challenge test. (**c**) WSSV titer measurement in indicated shrimp tissues.

**Figure 4 viruses-15-02010-f004:**
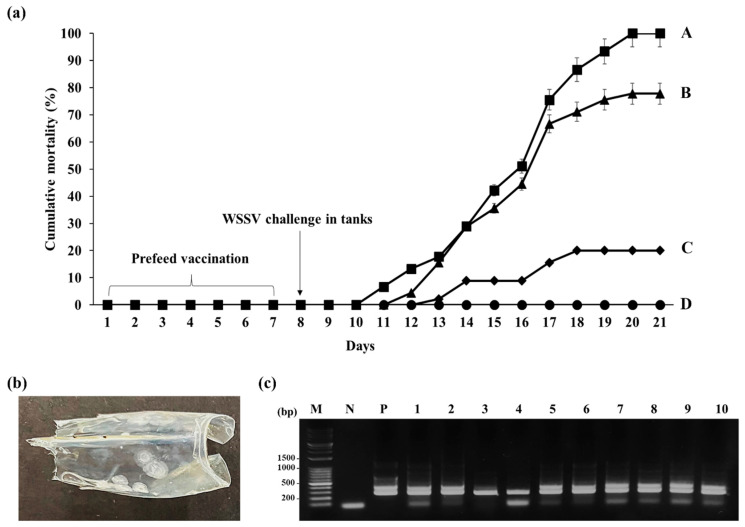
Cumulative mortality of *Litopenaeus vannamei* (*L*. *vannamei*) after oral immunization followed by a WSSV challenge. (**a**) The vaccination and challenge schedule and the cumulative mortality after the challenge. (●) Nonvaccinated, nonchallenged negative control. (■) Nonvaccinated, challenged positive control. (▲) The wild type *Chlorella vulgaris* (*C. vulgaris*)-fed group. (◆) The transformed *C. vulgaris*-fed group. Data are mean ± standard deviation of three replicates. The letters indicate significant differences (*p* < 0.001). (**b**) A carapace of a dead *L*. *vannamei* following challenge. (**c**) WSSV infection was confirmed by nested PCR analysis. Lane M: DM3200 DNA marker (SMOBIO); lane N: negative control; lane P: total DNA from *L*. *vannamei* used as positive control inoculum; lanes 1–10: total DNA from *L*. *vannamei* that died after challenge. The top and bottom bands in the positive control are primary and nested PCR products, respectively. The band in the negative control lane is primer DNA.

**Figure 5 viruses-15-02010-f005:**
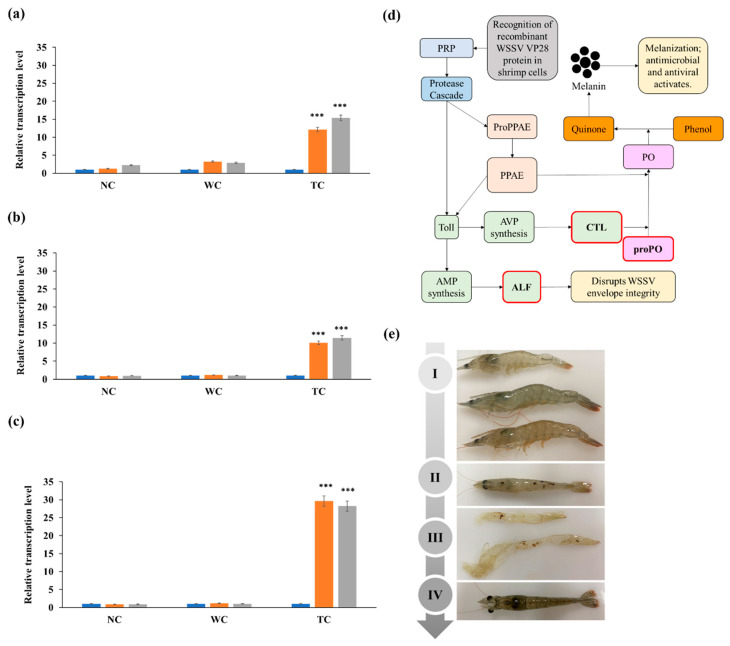
The transcription levels of immune-related genes following oral vaccination. (**a**–**c**) RT-qPCR analysis of the mRNA levels of the anti-lipopolysaccharide factor (ALF), C-type lectin (CTL), and prophenoloxidase (proPO) genes in *Litopenaeus vannamei* on days 0 (blue bars), 7 (orange bars), and 14 (gray bars), after vaccination with VP28-expressing *Chlorella vulgaris*. Data are mean ± standard deviation of three replicates. *** indicates significant differences at *p* < 0.001. (**d**) Shrimp immune pathways. (**e**) Changes in shrimp (*Litopenaeus vannamei*) after anti-WSSV oral vaccination. I: healthy shrimp; II: melanization; III: cuticular excretion; IV: shrimp with recovered health.

**Table 1 viruses-15-02010-t001:** Sequences of the primers used in this study.

Target	Primer Name	Sequences	PCR Product
1-NR	1-NR F	5′- ATGGACAAGACAGGGTTCGG -3′	2556 bp
	1-NR R	5′- CAACGAGCGCCTCCATTTAC -3′	
2-NR	2-NR F	5′- CACGAGGAGCATCGTGGAAA -3	2065 bp
	2-NR R	5′- AATACAGGCGGAGCCCAAAC -3′	
VP28	VP28 F	5′- TCGCTACCACAATACGGTG -3′	483 bp
	VP28 R	5′- TTGCCACCGGCTGTTG -3′	
WSSV	WSSV F	5′- CTTTCACTCTTTCGGTCGTGTC -3′	604 bp
	WSSV R	5′- TACTCGGTCTCAGTGCCAGA -3′	
Nested- WSSV	Nested F	5′- CCCACACAGACAATATCGAGACAA -3′	258 bp
	Nested R	5′- CTTGATGTGTTGTTCCACACCTTG -3′	
qWSSV	qWSSV F	5′- ATCCTCGCCATCACTGCTGT -3′	560 bp
	qWSSV R	5′- GAGTAGGTGACGTGCACGTA -3′	
qVP28	qVP28 F	5′- GTGACCAAGACCATCGAAAC –3′	110 bp
	qVP28 R	5′- TCAGTCATCTTGAAGTAGCC -3′	
ALF	ALF F	5′- CCTCATCCCTTCGCTAGTCCA -3′	148 bp
	ALF R	5′- CCATCCAGGACACCACATCC -3′	
Lectin	Lectin F	5′- TTACTCTCGGTGTCGTTGGGAC -3′	148 bp
	Lectin R	5′- ATCTGATACCAGGTTCCCTCAGTC -3′	
proPO	proPO F	5′- TATGAAGACGAACGGGGCC -3′	133 bp
	proPO R	5′- CAGACGGACGGAATGGAGTT -3′	
β-actin	β-actin F	5′- ATGTTCGAGACCTTCAACACCC -3′	135 bp
	β-actin R	5′- CTCGTAGATGGGCACGGT -3′	

## Data Availability

The data presented in this study are available on request from the corresponding author.

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
