# Peer review of "Efficacy of White Spot Syndrome Virus Protein VP28-Expressing Chlorella vulgaris as an Oral Vaccine for Shrimp"

_viruses, 2023, doi:10.3390/v15102010_

Round 1

Reviewer 1 Report

This is a straight forward feeding experiment and infection experiment conducted to evaluate the efficacy of Efficacy of White Spot Syndrome Virus Protein VP28-Expressing Chlorella vulgaris as an Oral Vaccine for Shrimp.The manuscript provides interesting and valuable data. The results of this study are beneficial to the development of healthy culture for shrimp. However, there are some questions which need careful explanation.

Q1. The manuscript pointed out that the VP19 and VP28 play critical roles during infection,Why did you choose VP28 instead of VP19? Does VP19  have the same effect?

Q2. In Figure 2d, it said TF1 expressed the highest level, whether there is an actin protein for reference?

Q3. In the experimental design, lack of the group that only transfected with empty plasmid pCCVG, instead of using wild-type Chlorella-fed as control, we can not determine whether VP28 protein or pCCVG reduced shrimp mortality.

Q4.  How does VP28 withstand temperature? Whether it will cause the loss of activity during the production of shrimp feed?

Q5. Is the white spot virus protein VP28 expressed by Chlorella vulgaris deactivated by protease in shrimp intestine?

Q6. How to use the White Spot Syndrome Virus Protein VP28-Expressing products properly? 

The English voice and tense in this manuscript are not expressed accurately enough in some sentences, and it is suggested to revise, for example,In this study, we developed and evaluated an oral WSSV vaccine using recombinant Chlorella that expressed WSSV’s VP28.It is suggested that the authors should change the sentence of"In this study, we developed and evaluated an oral WSSV vaccine using recombinant Chlorella that expressed WSSV’s VP28" into "In this study,  oral WSSV vaccine using recombinant Chlorella that expressed WSSV’s VP28 was developed and evaluated"

Author Response

This is a straight forward feeding experiment and infection experiment conducted to evaluate the efficacy of Efficacy of White Spot Syndrome Virus Protein VP28-Expressing Chlorella vulgaris as an Oral Vaccine for Shrimp.The manuscript provides interesting and valuable data. The results of this study are beneficial to the development of healthy culture for shrimp. However, there are some questions which need careful explanation.

Q1. The manuscript pointed out that the VP19 and VP28 play critical roles during infection,Why did you choose VP28 instead of VP19? Does VP19 have the same effect?

  • Both VP28 and VP19 of WSSV are known to exhibit protective efficacy against WSSV. After reviewing papers from the past 5 years, there were some papers showing higher defense efficiency in VP28 than VP19, so we selected VP28 and conducted the study.

Q2. In Figure 2d, it said TF1 expressed the highest level, whether there is an actin protein for reference?

  • There is no l antibody against actin protein that is commercially Therefore, the western blot was performed by using the same amount of total protein from all transformd Chlorella by quantification of extracted total protein by Braford Assay. WSSV virus was used as a positive control.

Q3. In the experimental design, lack of the group that only transfected with empty plasmid pCCVG, instead of using wild-type Chlorella-fed as control, we can not determine whether VP28 protein or pCCVG reduced shrimp mortality.

  • In the transformation assay, only DNA encompassing the 5’ NR fragemenet-35S promoter-terminator-3’ NR fragment was amplified by PCR and used for transformation. The 5’ and 5’ fragment of the NR (nitrate reductase) gene fragment was used the incorporation by homologous recombination, which allows the selection of transformed chlorella on a plate containing KClO3. This also allow transformation of Chlorella without the introduction of an antibiotic resistance gene. The methods is explained in detail in the cited reference 17. For this reason, we used wild type Chlorella for comparison.

Q4.  How does VP28 withstand temperature? Whether it will cause the loss of activity during the production of shrimp feed?

  • For feeding process, freeze-dried transgenic chlorella (chlorella that expresses WSSV VP28) was mixed with a commercial eel feed which made the feed like flour dough, which is described in the section 2.7 the Material and Methods. So, there was no high temperature treatment during feed preparation. and kneading it before use.

Q5. Is the white spot virus protein VP28 expressed by Chlorella vulgaris deactivated by protease in shrimp intestine?

  • There is some possibility of degradation in shrimp intestine. However, intact protein is not required for immune response because small portion of antigenic protein is required to elicit immune response. We are going to compare the efficacy of intact and sonicated Chlorella in future experiments to evaluate the rigid Chlorella cell wall as a natural encapsulating agent.

Q6. How to use the White Spot Syndrome Virus Protein VP28-Expressing products properly? 

  • Chlorella is listed as GRAS, a generally stable substance, and is widely used as a feed additive in fish farms because it is rich in nutrients. Our data should that freeze drying can completely inactivate Chlorella. Therefore, a vaccine feed could be prepared by mixing the e inactivated chlorella with normal feed materials. In this case, we need to test the temperature sensitivity of VP28 expressed in transformed Chlorella. Or, the inactivated chlorella could be mixed with other feed materials without heating as described here. In either case, evaluation of biosafety for accidental release of the transformed Chlorella into nature should be evaluated, which is planned in Korea.

The English voice and tense in this manuscript are not expressed accurately enough in some sentences, and it is suggested to revise, for example,In this study, we developed and evaluated an oral WSSV vaccine using recombinant Chlorella that expressed WSSV’s VP28.It is suggested that the authors should change the sentence of"In this study, we developed and evaluated an oral WSSV vaccine using recombinant Chlorella that expressed WSSV’s VP28" into "In this study,  oral WSSV vaccine using recombinant Chlorella that expressed WSSV’s VP28 was developed and evaluated"

  • Changed as suggested

Reviewer 2 Report

The manuscript entitled “Efficacy of White Spot Syndrome Virus Protein VP28-Expressing Chlorella vulgaris as an Oral Vaccine for Shrimp” presented original and hot-topic data and it was designed very well. In addition, the manuscript has a good English style. The Authors synthesized a DNA fragment conferring an efficient survival rate against the white spot syndrome virus in Shrimps. However, there are some concerns listed below which should be considered before publication:

1-Please add some notes about the importance of edible vaccines compared to the other kinds of vaccines.

2-Please add some research done on some microalgae for the production of edible vaccines. You can use the following article (https://doi.org/10.3390/md20110657).

3-Generally, the efficiency of vaccines is determined by ELISA assay and some immune markers such as IL-2, CD4+, CD8+ T cells, and so on. Please explain more about the estimation of vaccine efficacy by RT-PCR approaches.  

4-Given you used animals in your research, please; add an ethics statement in the text if it is available.

English quality is very good. 

Author Response

The manuscript entitled “Efficacy of White Spot Syndrome Virus Protein VP28-Expressing Chlorella vulgaris as an Oral Vaccine for Shrimp” presented original and hot-topic data and it was designed very well. In addition, the manuscript has a good English style. The Authors synthesized a DNA fragment conferring an efficient survival rate against the white spot syndrome virus in Shrimps. However, there are some concerns listed below which should be considered before publication:

  • Please add some notes about the importance of edible vaccines compared to the other kinds of vaccines.
  • Thank you for your comments. Importance of edible vaccine considering the size of shrimp and large scale of shrimp farm is mentioned added on lines 51-53
  • Please add some research done on some microalgae for the production of edible vaccines. You can use the following article (https://doi.org/10.3390/md20110657).

=> This is mentioned on lines 59-64 with the citation of the recommended article.

3-Generally, the efficiency of vaccines is determined by ELISA assay and some immune markers such as IL-2, CD4+, CD8+ T cells, and so on. Please explain more about the estimation of vaccine efficacy by RT-PCR approaches.  

  • Thank you for your comment. The definite effect of a vaccine could be determined the relative percent survival rate (RPS) of 80%, which showed that the oral vaccine we used is effect. Up to now, there is no evidence of adaptive immunity such as CD4+, CD8+T cells that you mentioned in crustacean. Therefore, we investigated the expression levels of proPO, ALF, and CTL genes that are known to be involved in the innate immune response. The result of RT-PCR of these genes and their possible function in WSSV protection are shown in Fig. 5.
  1. Given you used animals in your research, please; add an ethics statement in the text if it is availabl.

=> I appreciate your concern. There is regulation about ethics regulation for invertebrates including crustaceans in Kora. Also, the authors guideline for “Viruses” requests the ethics statement for reports on research that involves human subjects, human material, human tissues, or human data,

Reviewer 3 Report

Paper is well interesting, shown a important finding to induce shrimp immunity.

However there is lack of indepth analysis to prove how VP28-expressing Chlorella vulgaris work as vaccine.

Like shrimp lack adaptive immunity so author needs to prove how VP28-expressing Chlorella vulgaris works, whether it was innate immune priming or other mechanism. DNA/RNA methylation, histone modifications, etc.

Several grammatical mistakes.  Thorough revision is required.

Author Response

Paper is well interesting, shown a important finding to induce shrimp immunity.

 However there is lack of indepth analysis to prove how VP28-expressing Chlorella vulgaris work as vaccine.

  • As shown in Fig 4, we have shown that compared to shrimp fed on wildtype vulgaris, which had a final cumulative mortality of 77.8%, shrimp fed on VP28-expressing C. vulgaris had a final cumulative mortality of 20% and a relative percent survival rate (RPS) of 80%, which shows definite efficacy of the WSSV VP28 expressing chlorella as an oral vaccine. Furthermore, we analyzed the expression levels of proPO, ALF, and CTL genes from shrimps fed with the WSSV VP28 expressing chlorella by RT-qPCR. These genes are known to be involved in the innate immune response in invertebrates. Based on the results, we proposed the possible mechanisms of protection in Fig. 5. All these data should be enough to show the efficacy of the WSSV VP28 expressing chlorella as an oral vaccine.

Like shrimp lack adaptive immunity so author needs to prove how VP28-expressing Chlorella vulgaris works, whether it was innate immune priming or other mechanism. DNA/RNA methylation, histone modifications, etc.

  • Thank you for your comments. As you commented these showed be many different mechanisms of protection after oral vaccination, which would be impossible to address in one manuscript. As mentioned above, we analyzed the expression levels of proPO, ALF, and CTL genes from shrimps fed with the WSSV VP28 expressing chlorella by RT-qPCR. These genes are known to be involved in the innate immune response in invertebrates. Although these results can not explain the whole events in the vaccinated shrimp, they gave an idea of protection mechanism as shown in Fig. 5.

Comments on the Quality of English Language: Several grammatical mistakes.  Thorough revision is required.

  • We used two different English Editing Company before submission of the manuscript, as shown below and checked before submission and hope this can be answer to your concern.

Round 2

Reviewer 1 Report

In the experimental design, lack of the group that only transfected with empty plasmid pCCVG, instead of using wild-type Chlorella-fed as control, we can not determine whether VP28 protein or pCCVG reduced shrimp mortality.Although this does not affect the publication of the paper, the experimental design would have been more perfect if this treatment had been  supplied.

The English expression of the revised manuscript  has been improved to some extent

Reviewer 3 Report

Paper can be accepted for publication. 

Author Response

Thank you for your review and kind suggestions.